# Improving Autoregressive Image Generation by Mitigating Gradient Bias in Softmax

## Abstract

Softmax is the most commonly used probabilistic activation function in classification tasks, partly due to its tendency to over-penalize non-target classes with high prediction scores. However, this property becomes detrimental in autoregressive generation tasks, where multiple valid predictions may exist. Unlike conventional classification task, which seeks a single correct answer, autoregressive models are expected to assign high probabilities to various plausible outputs to ensure diversity in generation. However, during training, gradient bias caused by Softmax over-penalizes non-target predictions with high probabilities, limiting output diversity and hindering optimization convergence. To alleviate this, we propose Gradient Suppressed Softmax (GS-Softmax), which reduces the gradient contributions of high-probability non-target classes. Through experiments, we demonstrate that GS-Softmax improves both the diversity of generated content and optimization convergence. Code and pre-trained models will be made public.

## 1 Introduction

Autoregressive models, leveraging the "predicting next token in the sequence" training strategy and transformer structure (Vaswani, 2017), have achieved remarkable success in language modeling (Radford, 2018; Brown, 2020; Touvron et al., 2023). Recently, building upon VQVAE (Van Den Oord et al., 2017), which represents images as sequences of discrete tokens, the "next token prediction" paradigm has also shown its effectiveness in image generation (Tian et al., 2024; Sun et al., 2024; Li et al., 2024), surpassing diffusion-based methods (Song & Ermon, 2019; Ho et al., 2020; Dhariwal & Nichol, 2021; Lu et al., 2022; Rombach et al., 2022) on several benchmarks.

The next token prediction task in autoregressive generative models is a specialized form of classification, where the model learns to predict the probability distribution of the next token based on the preceding tokens in a sequence. Unlike conventional classification tasks that seek a single correct answer, these models should assign high probabilities to multiple plausible outputs to ensure both reasonable and diverse generations. For instance, given the sequence "His name is," the model should favor various possible names rather than one.

However, autoregressive generative models are typically trained in a self-supervised manner (Radford, 2018; Brown, 2020; Touvron et al., 2023; Sun et al., 2024; Tian et al., 2024), where the model predicts a single target token during training and is optimized to maximize the probability of this label. This suppresses other valid predictions, leading to overconfidence in one outcome. We find that this issue is further exacerbated by the use of Softmax, which amplifies the focus on the predicted target while diminishing diversity in the output.

Probabilistic activation functions (PAF) are essential in machine learning for converting model outputs (*i.e.*, logits) into probability distributions. Among them, Softmax (Bridle, 1989) has emerged as the dominant choice, particularly in classification tasks, owing to its propensity to excessively penalize high probabilities for non-target classes. This makes Softmax highly effective for single-label classification problems. However, in autoregressive generation tasks, where multiple valid options often exist, Softmax can become problematic. In these cases, with Softmax, the optimization will focus more on suppressing potentially valid but non-target predictions, as they naturally carry high probabilities (Holtzman et al., 2019), rather than maintaining a balanced distribution across all plausible options. This misalignment not only limits the diversity of the model's generated content but also hinders its ability to properly learn the underlying data distribution.

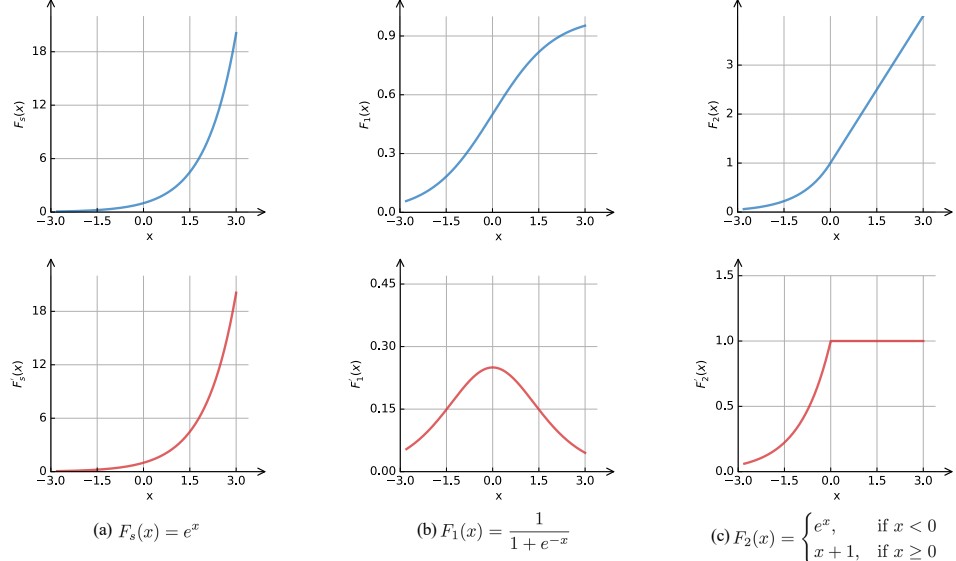

(a) $F_s(x) = e^x$

(b) $F_1(x) = \dfrac{1}{1 + e^{-x}}$

(c) $F_2(x) = \begin{cases} e^x, & \text{if } x < 0 \\ x + 1, & \text{if } x \geq 0 \end{cases}$

Figure 1: Visualization of the positive mapping functions (indicated with blue) and their derivative (indicated with red). When $x$ and $F(x)$ are large, the derivatives of our functions (b, c) stop increasing, while the derivative of the mapping function (a) used in Softmax continues to increase.

To address the misalignment, we need to solve the issue where Softmax overly penalizes high-probability non-target predictions. The Softmax operation can generally be broken down into two steps: (1) applying a positive mapping function (PMF) to transform model outputs (*i.e.*, logits) into positive values, and (2) normalizing these values so that their sum equals one. By default, Softmax uses $F_s(x) = e^x$ as its PMF, whose derivative increases with the input. This results in larger gradients flow through logits with higher values, leading to the over-penalization of potential valid predictions. To address this gradient bias, we propose to replace $F_s$ with a function whose derivative does not continuously grow as the input increases. Based on this criteria, we design two positive mapping functions $F_1$ and $F_2$ (see illustration in Figure 1).

Through experiments, we demonstrate that replacing the PMF in Softmax with $F_1$ or $F_2$ leads to performance improvements, validating our analysis and findings. We refer to this variant as Gradient Suppressed Softmax (GS-Softmax). In a series of comprehensive experiments, we further show that GS-Softmaxnot not only enhances the diversity of generated content but also improves optimization convergence compared to traditional Softmax.

## 2 PRELIMINARY

### 2.1 PROBABILISTIC ACTIVATION FUNCTION

In classification tasks, models are required to predict the likelihood of each class. To achieve this, classification models usually use a probabilistic activation function (PAF) to convert the model outputs (*i.e.*, logits) into a valid probability distribution (*i.e.*, all values are positive and add up to one). Typically, the probabilistic activation functions can be conceptualized as a two-step process:

1. Positive Mapping Function (PMF): Use a positive monotonically increasing function $F$ (*e.g.*, $F_s(x) = e^x$ in Softmax) to convert the logits into positive values.

2. Normalization: Normalize the positively mapped values so that their sum equals one.

Let $K$ represent the number of classes in the task. The calculation of Softmax (Bridle, 1989), a widely used probability activation function, can be expressed as:

$$\text{Softmax}(x^{(i)}) = \frac{F_s(x_i)}{\sum_{j=1}^{K} F_s(x_j)}, \quad \text{where } F_s(x) = e^x. \tag{1}$$

## 2.2 AUTOREGRESSIVE MODEL

Autoregressive (AR) models are developed based on the chain rule of probability, which allows the joint probability of a set of variables $x = (x_1, x_2, \ldots, x_n)$ to be factorized into a product of conditional probabilities for each component:

$$p(x) = p(x_1, \ldots, x_n) = \prod_{i=1}^{n} p(x_i \mid x_1, \ldots, x_{i-1}). \tag{2}$$

To learn the data distribution $P$, most autoregressive generative models are trained based on the "next token prediction" strategy. Specifically, let $(x_1, x_2, \ldots, x_t)$ denote a token sequence sampled from the dataset, where the goal is to predict $x_t$ based on the preceding tokens $(x_1, x_2, \ldots, x_{t-1})$. This is achieved by maximizing the predicted probability $Q_\theta(x_t|x_{<t})$ using maximum likelihood estimation (MLE). Inherently, the training objective of MLE is equivalent to minimizing the cross-entropy between true data distribution $P$ and the model's predicted distribution $Q_\theta$, which can be formulated as:

$$\mathcal{L}_{\text{MLE}} = -\mathbb{E}_{x \sim P}[\sum_{t=1}^{|x|} \log Q_\theta(x_t|x_{<t})]. \tag{3}$$

## 3 METHOD

Most autoregressive image generation models (Yu et al., 2021; Esser et al., 2021; Lee et al., 2022; Sun et al., 2024; Tian et al., 2024) adopt Softmax as the probabilistic activation function. In this section, we first demonstrate that using Softmax inherently conflicts with the training objective of autoregressive generative models. Then, to alleviate this issue, we introduce a novel probabilistic activation function: Gradient Suppressed (GS) Softmax.

### 3.1 SOFTMAX IS NOT OPTIMAL FOR AUTOREGRESSIVE GENERATIVE MODEL

In autoregressive tasks such as text or image generation, a single input can have multiple plausible continuations. For instance, the prompt "My name is" could be followed by various valid completions. Thus, the autoregressive generative model should be capable of assigning high probabilities to various reasonable answers, as this is essential for implementing sampling strategies such as top-$k$ (Fan et al., 2018) and top-$p$ sampling (Holtzman et al., 2019) during inference. These sampling strategies, in turn, are key to ensuring the model's generations are diverse and valid.

In this section, we first reveal that the training objective of autoregressive generative models contradicts the expectation that they should assign high probabilities to all potential correct answers. Then, we demonstrate that this issue is further exacerbated when Softmax is used.

### 3.1.1 CONTRADICTION BETWEEN TRAINING OBJECTIVE AND EXPECTATION

Since autoregressive models are typically trained in a self-supervised manner, each training sample is paired with only one "correct" target output, which is the next token in the given sequence. As proved below, this forces the model to reduce the predicted probabilities of all other options.

For simplicity, let $L_\theta^T(x)$ and $Q_\theta^T(x)$ respectively denote logits and the probability assigned by the model to the target class $T$ given the input sequence $\{x_1, x_2, \ldots, x_{t-1}\}$, while $L_\theta^{N_i}(x)$ and $Q_\theta^{N_i}(x)$ denote the ones assigned to the $i$-th non-target class $N_i$. Then the calculation of $Q_\theta^T(x)$ can be termed as:

$$Q_\theta^T(x) = \frac{F(L_\theta^T(x))}{F(L_\theta^T(x)) + \sum_{i=1}^{K-1} F(L_\theta^{N_i}(x))}. \tag{4}$$

Then, the training loss calculated on this single training example can be termed as:

$$\mathcal{L}_{\text{MLE}}(x) = -\log(Q_\theta^T(x))$$
$$= \log(F(L_\theta^T(x)) + \sum_{i=1}^{K-1} F(L_\theta^{N_i}(x))) - \log(F(L_\theta^T(x))). \tag{5}$$

As can be inferred from equation 5, and considering $F$ is a monotonically increasing function, minimizing $\mathcal{L}_{\text{MLE}}(x)$ forces the logits of all non-target classes to decrease. Accordingly, their predicted probabilities are decreased, even if some of them could also be valid predictions in autoregressive generation missions.

### 3.1.2 SOFTMAX OVER-PENALIZES POTENTIAL CORRECT PREDICTIONS.

Ideally, during training, the penalization of potentially valid non-target predictions should be minimized to prevent excessively lowering their likelihood. However, with Softmax, this penalization is actually exacerbated. According to equation 5, the gradient calculation for non-target logits $L_\theta^{N_i}(x)$ can be referred to as follows:

$$\frac{\partial \mathcal{L}_{\text{MLE}}(x)}{\partial L_\theta^{N_i}(x)} = c \times \frac{\partial F(L_\theta^{N_i}(x))}{\partial L_\theta^{N_i}(x)}, \quad \text{where } c = \frac{1}{F(L_\theta^{T}(x)) + \sum_{i=1}^{K-1} F(L_\theta^{N_i}(x))}. \tag{6}$$

According to equation 6, among non-target classes, the relative magnitudes of the gradients flow to their logits are governed by $\frac{\partial F(L_\theta^{N_i}(x))}{\partial L_\theta^{N_i}(x)}$, considering $c$ is identical across them. Since Softmax uses the exponential function $F_s(x) = e^x$ as the positive mapping function, and its derivative grows as $x$ increases, higher logits receive larger gradients. As a result, non-target classes with higher predicted probabilities, which are always potentially correct predictions in autoregressive generation tasks (Fan et al., 2018; Holtzman et al., 2019), experience stronger suppression due to larger gradient updates. This contradicts the expectation that autoregressive generative models should assign high probabilities to as many potential valid candidates as possible, as discussed in Section 3.1.

### 3.2 GRADIENT SUPPRESSED SOFTMAX

As analyzed in Section 3.1.2, the over-penalization problem arises since Softmax will assign greater gradients to higher logits. Inherently, this gradient bias arises since $F_s$ is a monotonically increasing function. Based on this recognition, we propose to replace $F_s$ with the function $F$ that meets the following criteria:

1. $F'(x)$ should not increase as $F(x)$ increases when $F(x)$ is large.
2. The function $F(x)$ should be positive and monotonically increasing ($i.e.$, $\forall x_2 > x_1, F(x_2) > F(x_1) > 0$).
3. $\lim_{F(x) \to 0} F'(x) = 0$.

Among them, criterion 1 is used to alleviate the over-penalization problem by ensuring that larger logits do not receive greater gradients. Criteria 2 and 3 draw on the advantages of Softmax, ensuring the efficiency and stability of the training. Considering these criteria and computational efficiency, we design the following two functions that meet the above three requirements:

$$\text{Function 1:} \quad F_1(x) = \frac{1}{1 + e^{-x}} \tag{7}$$

$$\text{Function 2:} \quad F_2(x) = \begin{cases} e^x, & \text{if } x < 0 \\ x + 1, & \text{if } x \geq 0 \end{cases} \tag{8}$$

Both Function 1 and Function 2 meet the requirements above (see the visualization in Figure 1). By default, we use Function 1 as the positive mapping function, while we also empirically verify the effectiveness of Function 2 in Section 5.1. Then, the calculation of GS-Softmax can be termed as:

$$\text{GS-Softmax}(x_i) = \frac{F_1(x_i)}{\sum_{j=1}^{K} F_1(x_j)}, \quad \text{where } F_1(x) = \frac{1}{1 + e^{-x}}. \tag{9}$$

### 3.3 TEMPERATURE SCALING IN GS-SOFTMAX

In the inference stage of autoregressive generation tasks, temperature scaling (Hinton, 2015; Guo et al., 2017) can be employed to modulate the sharpness of probability distributions, which allows

for trade-offs between the diversity and quality of the generated content. For Softmax, considering its positive mapping function $F_s(x) = e^x$ is a convex function, the temperature scaling is realized by simply scaling the input logits with a factor $1/\tau$, where $\tau$ is the temperature parameter.

Since the positive mapping function used in GS-Softmax is non-convex, we propose to perform temperature scaling on "fake logits". Let $P = \{p_1, p_2, \ldots, p_k\}$ denote the probability distribution of the next token predicted by the model, we use first use $M(x) = \ln(x)$ to map the predicted probabilities to "fake logits" $L^F = \{\ln(p_1), \ln(p_2), \ldots, \ln(p_k)\}$. As proved below, the "fake logits" can be converted back to $P$ by applying Softmax:

$$
\begin{aligned}
\text{Softmax}(L_i^F) &= \frac{e^{\ln(p_i)}}{\sum_{j=1}^{K} e^{\ln(p_j)}} \\
&= \frac{p_i}{\sum_{j=1}^{K} p_j} \\
&= p_i.
\end{aligned}
\tag{10}
$$

Considering now that the probabilities are obtained with "fake logits" and Softmax, we can perform temperature scaling on the "fake logits" like the original algorithm (Hinton, 2015). Then, in GS-Softmax, the calculation of the scaled probability $p_i^\tau$ can be termed as:

$$
p_i^\tau = \frac{e^{\ln(p_i)/\tau}}{\sum_{j=1}^{K} e^{\ln(p_j)/\tau}}.
\tag{11}
$$

## 4 EXPERIMENTS

### 4.1 SETTINGS

To validate the effectiveness of GS-Softmax, we conduct experiments following the setup of LLa-maGen (Sun et al., 2024), which achieves state-of-the-art performance with a straightforward algorithmic design. We outline the training setup below and refer the reviewer to (Sun et al., 2024) for further details.

**Training.** We train models using Softmax and GS-Softmax on the widely used 256×256 ImageNet benchmark using consistent settings. Specifically, for all experiments, the learning rate is set as $10^{-4}$ with a batch size of 256, and we use the AdamW optimizer with $\beta_1 = 0.9$, $\beta_2 = 0.95$, and a weight decay of 0.05. All models are trained for 300 epochs. In addition, gradient clipping with a maximum norm of 1.0 is utilized. The dropout rate is 0.1 for input token embeddings, attention layers, and feedforward networks (FFN). We also apply a 0.1 dropout to the class condition embedding for classifier-free guidance.

**Evaluation.** We evaluate the effectiveness of our method using Fréchet Inception Distance (FID) (Heusel et al., 2017) and sFID (Nash et al., 2021) as the primary metrics, as they consider both the diversity and quality of the generated images. Additionally, we also report Inception Score (IS) (Salimans et al., 2016) and Recall (Kynkäänniemi et al., 2019) for a comprehensive comparison. For fairness, we fix the random seed as 0 and use the evaluation scripts provided by Sun et al. (2024). Without special specification, all experiments in Section 4 are performed on ImageNet with $256 \times 256$ resolution using GPT-B. By default, as suggested by Sun et al. (2024), we set CFG=2, temperature=1, and use top-k sampling to perform the evaluation.

### 4.2 MAIN RESULTS

To evaluate the effectiveness of GS-Softmax, we train models using either Softmax or GS-Softmax, assessing their performance based on the following two aspects: the convergence of optimization and the generative capacity of the trained models.

**Effect on optimization convergence.** Theoretically, using GS Softmax might increase training loss since it suppresses the penalty on high-confidence, non-target predictions. However, in practice, we find GS Softmax always helps to reduce the loss on both training and validation datasets. In Table 1, we compare the losses of models using standard Softmax versus GS-Softmax, where perplexity

| Model | GPT-B | | GPT-L | | GPT-XL | |
|---|---|---|---|---|---|---|
| PAF | Softmax | GS-Softmax | Softmax | GS-Softmax | Softmax | GS-Softmax |
| $PPL_{train}$ | 1954.84 | **1949.62** | 1483.53 | **1473.65** | 1221.89 | **1208.16** |
| $PPL_{val}$ | 2021.07 | **2018.46** | 1644.99 | **1634.54** | 1553.13 | **1548.72** |

Table 1: Perplexity (PPL, equal to $e^{\mathcal{L}_{MLE}}$, lower is better) of the trained models on the training and validation splits of ImageNet. GS-Softmax helps the optimization converge better.

| Model | Top-$k$ | PAF | sFID ↓ | FID ↓ | Recall (%) ↑ | IS ↑ |
|---|---|---|---|---|---|---|
| GPT-B (111M) | Top-1000 | Softmax | 29.51 | 10.50 | 31.53 | 184.68 |
| | | GS-Softmax | **27.39** | **10.47** | **32.51** | **189.99** |
| | Top-5000 | Softmax | 11.31 | 6.73 | 39.76 | 203.00 |
| | | GS-Softmax | **10.53** | **6.69** | **40.91** | **205.17** |
| | Top-all | Softmax | 7.53 | 5.25 | 45.08 | 189.28 |
| | | GS-Softmax | **7.21** | **5.22** | **45.34** | **193.38** |
| GPT-L (343M) | Top-1000 | Softmax | 23.70 | 8.35 | 36.69 | **285.25** |
| | | GS-Softmax | **22.81** | **8.02** | **38.78** | 275.65 |
| | Top-5000 | Softmax | 10.32 | 4.95 | 46.27 | **288.17** |
| | | GS-Softmax | **9.39** | **4.66** | **47.58** | 280.50 |
| | Top-all | Softmax | 7.29 | 3.34 | 51.98 | **270.47** |
| | | GS-Softmax | **6.72** | **3.20** | **52.77** | 260.06 |
| GPT-XL (775M) | Top-1000 | Softmax | 19.58 | 7.11 | 39.20 | **322.05** |
| | | GS-Softmax | **19.36** | **6.76** | **40.75** | 308.15 |
| | Top-5000 | Softmax | 9.14 | 4.48 | 48.23 | **315.58** |
| | | GS-Softmax | **9.01** | **4.12** | **49.78** | 306.78 |
| | Top-all | Softmax | 6.95 | 3.09 | 52.84 | **300.40** |
| | | GS-Softmax | **6.88** | **2.96** | **53.56** | 285.57 |

Table 2: Performance comparisons on class-conditional ImageNet benchmark with 256×256 resolution. Better results are highlighted in **bold**.

(*i.e.*, $e^{\mathcal{L}_{MLE}}$) are reported for easier observation. The results show that, in most cases, using Gradient Suppressed Softmax can lower the models' perplexity on both training and validation sets. Given that the training objective is the same, lower perplexity suggests that GS-Softmax enables the model to learn the data distribution more effectively, as the optimization converges better. This further supports our analysis that over-penalizing potentially valid non-target predictions conflicts with the training objectives of autoregressive generative models.

**Effect on image generation diversity and quality.** We assess the quality of images generated under various settings. As shown in Table 2, compared with Softmax, using GS-Softmax continuously achieves better FID and sFID scores on different settings. This demonstrates that, with GS Softmax, models could maintain a more balanced prediction distribution across plausible options. Moreover, as can be observed, as the value of top-$k$ decreased, the improvement brought by GS Softmax consistently increased. This reflects that GS-Softmax helps the model assign higher probabilities to more "correct" candidates than Softmax does.

### 4.3 ABLATION

**Temperature scaling.** To verify the effectiveness of the temperature scaling algorithm we designed in Section 3.3, we evaluate our trained model with different temperature factors. The results in Table 3 show that as the temperature increases, the sFID, FID, and Recall scores continuously improve,

| Temperature | sFID ↓ | FID ↓ | Recall (%) ↑ | Inception↑ | Precision (%) ↑ |
|---|---|---|---|---|---|
| 0.98 | 7.55 | 5.43 | 44.45 | **196.91** | **84.82** |
| 0.99 | 7.41 | 5.35 | 45.21 | 195.13 | 84.73 |
| 1.0 | 7.21 | 5.22 | 45.34 | 193.38 | 84.55 |
| 1.01 | 7.17 | 5.20 | 45.71 | 192.30 | 84.54 |
| 1.02 | **7.15** | **5.16** | **46.21** | 190.30 | 83.90 |

Table 3: Performance comparisons when performing temperature scaling in the inference stage. GS-Softmax is employed as the probability activation function.

| Scale Factor $\alpha$ | sFID ↓ | FID ↓ | Recall (%) ↑ | Inception↑ | PPL$_{\text{train}}$ ↓ | PPL$_{\text{val}}$ ↓ |
|---|---|---|---|---|---|---|
| 0.5 | 7.57 | **5.03** | **47.34** | 191.97 | **1946.91** | **2016.21** |
| 1.0 | 7.21 | 5.22 | 45.34 | **193.38** | 1949.62 | 2018.46 |
| 2.0 | **7.18** | 5.24 | 46.95 | 184.76 | 1955.89 | 2023.45 |

Table 4: Ablation study on scaling the range of $F_1$ by a factor of $a$, where the range is enlarged or reduced by $a$ times. Enlarging the range degrades overall performance.

indicating the diversity of the generated outputs is enhanced. Conversely, when the temperature decreases, both the inception and precision scores improve, suggesting higher quality in the generated images. This verifies the effectiveness of our temperature scaling algorithm, showing that we can still use it to make a trade-off between diversity and quality when using GS-Softmax.

**Range of the positive mapping function.** The expressive power of the positive mapping function $F_1$ used in GS-Softmax is relatively weaker than that of the function used in Softmax (*i.e.*, $F_s(x) = e^x$), as the range of $F_s$ is $(0, +\infty)$, while the range of $F_1$ is $(0, 1)$. To explore whether this affects performance, we trained and evaluated models using GS-Softmax, scaling the range of $F_1$ by multiplying it by a factor $\alpha$. As shown in Table 4, increasing the range worsened the optimization convergence. We attribute this to the fact that expanding the range reduces the gradient-suppressing effect of GS-Softmax. Additionally, since carefully tuning the range provides minimal benefit, for the sake of simplicity, we set $\alpha = 1$ as the default.

**Classifier Free Guidance.** We conduct evaluations using different weights for Classifier-Free Guidance (CFG) to examine its impact. As can be observed in Table 5 (Left), a larger CFG weight results in better IS scores. Given that we consider sFID and FID as the primary metrics for evaluation, we set CFG=2 as the default configuration, which performs well across various settings.

## 5 DISCUSSION

### 5.1 SELECTION OF POSITIVE MAPPING FUNCTION

As discussed in Section 3.2, the derivative of the positive mapping function $F(x)$ should not continuously increase as $F(x)$ increases. Based on this, we introduce two alternative positive mapping functions: $F_1(x)$ (equation 7) and $F_2(x)$ (equation 8). The key difference between them is that $F_1'(x)$ decreases as $x$ increases for $x \geq 0$, while $F_2'(x) = 1$ for $x \geq 0$. Both of them have lower derivatives when $x$ is large compared to the mapping function $F_s$ used in Softmax (equation 1), whose derivative keeps growing as $x$ increases.

To validate the effectiveness of the criteria we suggested in Section 3.2, we trained models using either $F_s$, $F_1(x)$ (equation 7) or $F_2(x)$ (equation 8) as the positive mapping functions and compared their performance. As shown in Table 6, both $F_1(x)$ and $F_2(x)$ outperform the original Softmax function. This confirms the effectiveness of our approach, demonstrating that over-penalizing potentially correct candidates harms performance. Additionally, $F_1$ and $F_2$ did not demonstrate a clear superiority over each other in the image evaluation metrics. However, $F_1$ continuously achieved better perplexity scores on both the training set and validation set, indicating it helps the optimization converge better. Therefore, we default to using $F_1(x)$ as the positive mapping function.

| CFG | sFID ↓ | FID ↓ | Recall (%) ↑ | IS ↑ |
|-----|--------|-------|--------------|------|
| 1.75 | **7.21** | 5.97 | 49.58 | 160.36 |
| 2.0 | **7.21** | **5.22** | **45.34** | 193.38 |
| 2.25 | 7.32 | 5.48 | 42.8 | **223.30** |

| Model | SoftMax | GS-Softmax |
|-------|---------|------------|
| GPT-B (111 M) | 0.72 | 0.74 |
| GPT-L (343 M) | 1.15 | 1.18 |
| GPT-XL (775 M) | 2.58 | 2.60 |

Table 5: **Left**: Ablation on the weight of Classifier Free Guidance (CFG), IS scores become better as the weight of CFG increases. **Right**: Comparisons of training cost (GPU seconds per iteration), where different probabilistic activation functions are used.

| Resolution | PAF | sFID ↓ | FID ↓ | Recall (%) ↑ | Inception↑ | PPL$_{train}$ ↓ | PPL$_{val}$ ↓ |
|------------|-----|--------|-------|--------------|-----------|---------------|-------------|
| | Softmax | 7.53 | 5.25 | 45.08 | 189.28 | 1954.84 | 2021.07 |
| 256×256 | GS-Softmax ($F_1$) | 7.21 | 5.22 | 45.34 | **193.38** | **1949.62** | **2018.46** |
| | GS-Softmax ($F_2$) | **7.02** | **5.12** | **46.28** | 186.18 | 1951.35 | 2019.66 |
| | Softmax | 7.74 | 5.85 | 45.50 | 170.35 | 1513.77 | 1630.10 |
| 384×384 | GS-Softmax ($F_1$) | **7.64** | **5.80** | **46.06** | 168.68 | **1503.95** | **1618.97** |
| | GS-Softmax ($F_2$) | 7.69 | 5.83 | 45.41 | **170.93** | 1507.68 | 1624.58 |

Table 6: Performance comparisons of using different positive mapping functions. Models with GS-Softmax ($F_1$) consistently achieve better perplexity scores.

## 5.2 OVER PENALIZATION AND OVER CONFIDENCE

In standard classification tasks, several methods have been proposed to prevent the model from becoming overly confident in its predictions. These approaches also help to reduce the penalty on non-target high-probability categories during optimization. In this section, we explore whether methods aimed at reducing model confidence, such as label smoothing (Szegedy et al., 2016) and temperature scaling (Hinton, 2015), can benefit autoregressive image generation models.

**Temperature scaling does not benefit in the training stage.** As shown in Table 7, incorporating temperature scaling during training did not lead to performance improvements. We attribute this to the fact that although temperature scaling can make the predicted probability distribution either flatten or sharper, it does not alter the relative magnitude of the gradients—i.e., a larger probability still results in a larger gradient. As a result, as long as softmax is used, optimization will continue to focus more on reducing the probability of high-confidence, non-target classes, which is misaligned with the objectives of autoregressive generative models, as discussed in Section 3.1.

**Label smoothing.** Label smoothing (Szegedy et al., 2016) addresses model overconfidence by modifying the labels of non-target classes from 0 to a small constant. However, as shown in Table 8, applying label smoothing leads to a noticeable increase in both training loss and a decline in image evaluation metrics. We hypothesize that this degradation arises from the increased optimization challenge introduced by label smoothing. Moreover, it can also be attributed to the fact that label smoothing does not fundamentally solve the problem. When label smoothing is applied, all non-target labels remain the same. In this case, when Softmax is utilized, the optimization still focuses more on minimizing the probabilities of potential valid but non-target options. Considering this, it may be more effective to adjust the distribution of probabilities for non-target classes based on their predicted probabilities.

## 5.3 TRAING COST AND EFFICIENCY

**Training Cost.** Compared to Softmax, GS-Softmax introduces a few additional calculations, because the positive mapping function has been changed from $F_s(x) = e^x$ to $F_1(x) = \frac{1}{1+e^{-x}}$. As the training cost reported in Table 5 (Right), this change does not significantly impact the overall training speed. For instance, training GPT-XL (775M) with Softmax takes approximately 2.58 GPU seconds per, while using GS-Softmax takes 2.60 GPU seconds on average. This represents an overall increase of just 0.7%. Also, part of this slight cost increase may be attributed to PyTorch's optimiza-

| PAF | sFID ↓ | FID ↓ | Recall (%) ↑ | Inception↑ | PPL$_{train}$ ↓ | PPL$_{val}$ ↓ |
|---|---|---|---|---|---|---|
| Softmax ($\tau$=0.8) | 7.70 | 5.27 | 45.30 | 188.42 | 2019.70 | 2072.13 |
| Softmax ($\tau$=1.0) | 7.53 | 5.25 | 45.08 | 189.28 | 1954.84 | 2021.07 |
| Softmax ($\tau$=1.2) | 7.62 | 5.46 | 45.21 | 178.82 | 2286.32 | 2341.59 |
| GS-Softmax | **7.21** | **5.22** | **45.34** | **193.38** | **1949.62** | **2018.46** |

Table 7: Performance comparisons when training models with different temperatures. Temperature scaling is not helpful in training.

| PAF | sFID ↓ | FID ↓ | Recall (%) ↑ | Inception↑ | PPL$_{train}$ ↓ | PPL$_{val}$ ↓ |
|---|---|---|---|---|---|---|
| Softmax | 7.53 | 5.25 | 45.08 | 189.28 | 1954.84 | 2021.07 |
| Softmax + Label Smoothing | 13.33 | 9.84 | **48.39** | 135.42 | 2171.45 | 2261.72 |
| GS-Softmax | **7.21** | **5.22** | 45.34 | **193.38** | **1949.62** | **2018.46** |

Table 8: Performance comparisons on class-conditional 256×256 ImageNet benchmark. Label Smoothing is not beneficial for training autoregressive generative model.

tions in the implementation of the Softmax. How to implement GS-Softmax in a more efficient way will be the focus of our future work.

**Training Efficiency.** Intuitively, using GS-Softmax may slow down the convergence speed during training since it suppresses the gradients used for punishing non-target classes with high probability. In practice, although models using GS-Softmax often exhibit higher training loss than those with standard Softmax in the early stages, the loss typically drops below that of models with Softmax after around 50 to 80 epochs. Considering that autoregressive image generation models typically require training for over 300 epochs, GS-Softmax does not hinder convergence speed overall.

### 5.4 VISUALIZATION

In Figure 2, we visualize images generated by models using either Softmax or GS-Softmax, where the sampling is restricted to candidates with the top-500 probabilities. Overall, GS-Softmax produces images with more detailed features, supporting the observation that models with GS-Softmax better capture the underlying data distribution (Table 1), and this advantage becomes even more pronounced when the sampling range is constrained (Table 2).

## 6 RELATED WORKS

**Autoregressive Image Generation.** Autoregressive models for image generation convert 2D images into 1D sequences of pixels or tokens, generating each RGB pixel or token in a predefined order. Early works (Van Den Oord et al., 2016; Van den Oord et al., 2016; Parmar et al., 2018; Chen et al., 2020) have demonstrated the effectiveness of autoregressive models on generating RGB pixels, achieving comparable performance compared to generative adversarial network (Goodfellow et al., 2014; Brock, 2018; Karras et al., 2019; Kang et al., 2023). Following works such as VQ-GAN (Esser et al., 2021; Lee et al., 2022) significantly improved performance by conducting autoregressive learning in the latent space of VQVAE (Van Den Oord et al., 2017; Razavi et al., 2019a), where images are represented as token sequences. VQVAE-2 (Razavi et al., 2019b) and RQ-Transformer (Lee et al., 2022) employ a similar approach but use a stacked VQVAE, which represents images through hierarchical codes in the latent space. Moreover, inspired by BERT (Devlin, 2018), masked prediction models (Chang et al., 2022; Yu et al., 2023a;a;b) enable predicting multiple tokens at each step, significantly reducing the deploying cost. More recently, autoregressive image generation methods (Sun et al., 2024; Tian et al., 2024; Li et al., 2024) have shown promising performance, excels methods based on diffusion (Song & Ermon, 2019; Ho et al., 2020; Dhariwal & Nichol, 2021; Lu et al., 2022; Rombach et al., 2022) on several benchmarks.

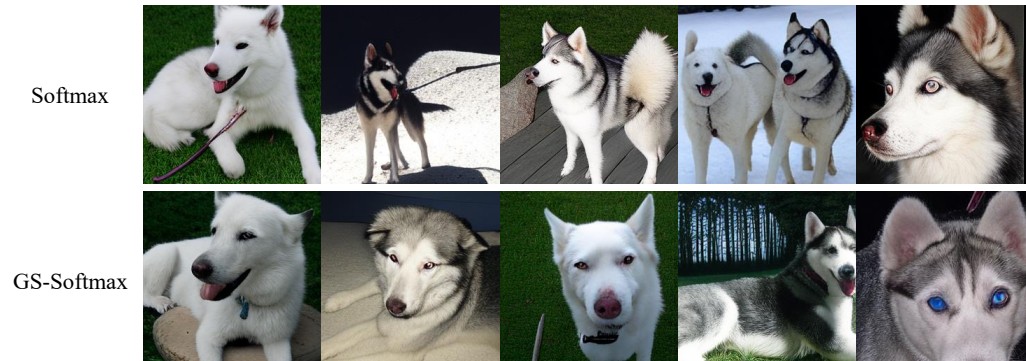

Figure 2: Non-cherry-picked Visualization of images generated by models using Softmax and GS-Softmax, conditioned on the class "Eskimo dog" (top-$k$=500). Images in the same column are generated with the same start token.

**Over-confidence issue in classification tasks.** In classification tasks, deep neural networks often exhibit an over-confidence problem, where the model assigns excessively high confidence to a single prediction while all others are greatly suppressed, which limits the generalizability of the trained model. To address this issue, label smoothing (Szegedy et al., 2016) smooths the hard labels of training samples to prevent the model from becoming overly confident in the target class. Temperature scaling (Guo et al., 2017) adjusts the temperature parameter of the softmax function to better calibrate the model's predicted probabilities, making them more reflective of actual confidence. Moreover, Pereyra et al. (2017) proposed to use both confidence penalty and label smoothing. In general, methods focusing on alleviating over-confidence problems often do not consider how to adjust confidence between non-target classes. Consequently, they are not effective in mitigating the over-penalization issue caused by Softmax in autoregressive generative model scenarios, as we have discussed and verified in Section 5.2.

**Variants of Softmax.** Various softmax variants have been proposed to enhance the performance. For example, Hierarchical Softmax (Morin & Bengio, 2005) and Taylor Softmax (Banerjee et al., 2020) are proposed to improve computing efficiency. Martins & Astudillo (2016) proposed Sparsemax, which produces sparse probability distributions, enabling models to focus on a subset of classes and improving interpretability. Moreover, Gumbel-Softmax (Jang et al., 2016) is proposed to facilitate differentiable sampling from categorical distributions, making it suitable for applications in variational inference. In this work, to alleviate the over-penalty problem caused by Softmax, we introduced a new variant: Gradient Suppressed (GS) Softmax.

## 7 CONCLUSION

In this work, we identify a key issue in autoregressive generative models: during training, the gradient bias introduced by Softmax will overly penalize potentially valid predictions, limiting the diversity of the generation and hindering the model's ability to learn the true data distribution. To address this, we propose to mitigate the gradient bias by replacing the exponential function in Softmax with a gradient-suppressed alternative. Through comprehensive experiments, we verify the effectiveness of this strategy and introduce Gradient Suppressed Softmax, which improves both generation diversity and training efficiency. We hope our work helps the following researches to design a more suitable probability activation function for autoregressive generative models.

**Limitation.** Intuitively, GS-Softmax should also be effective in autoregressive text generation tasks. However, we didn't verify this through experiments due to the limitation of the computing resource, considering it requires training multiple large language models from scratch.

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
