# OpenReview forum: "Improving Autoregressive Image Generation by Mitigating Gradient Bias in Softmax"
_ICLR.cc/2025/Conference — ICLR 2025 Conference Withdrawn Submission_

### Official Review · Reviewer_A9TR · 2024-10-31

**Soundness:** 3
**Presentation:** 3
**Contribution:** 2
**Rating:** 3
**Confidence:** 4

**Summary:**

This paper presents Gradient Suppressed Softmax for auto-regressive image generation.
The authors show that multiple valid tokens could be possible during generation and the traditional softmax compresses the probability of non-target tokens.
The authors then designed two types of new PMF for softmax operation, which shows slightly better FID for image generation.

**Strengths:**

* The studied problem is interesting. While auto-regressive image generation is popular, the sampling during inference is indeed an issue.
* Comprehensive ablation study of the proposed operations.

**Weaknesses:**

* The proposed method is more like a trick.
* The improvement is really marginal in terms of FID. On larger models, IS get decreased with the proposed method.
* Precision is missing from Table 2.

**Questions:**

See above

---

### Official Review · Reviewer_GVKs · 2024-11-02

**Soundness:** 3
**Presentation:** 3
**Contribution:** 3
**Rating:** 8
**Confidence:** 4

**Summary:**

This paper has the potential to question one of the most commonly used functions in deep learning.
It claims to introduce a variant of softmax that shows improved generation diversity and convergence, particularly for the autoregressive image models.
The proposed gradient-suppressed softmax learns potential candidates for pixels rather than obvious ones during training. It supports the claim by demonstrating improved diversity (tends to have better diversity key metrics like sFID, FID) and convergence (assuming autoregressive models will need training >300 epochs or so).

**Strengths:**

1. The problem is well understood and clearly articulated
2. Limitations are considered and argued
3. Supported by detailed experiments

**Weaknesses:**

1. Is this function scalable for higher-context windows? (or as shown fixed for the 256x256, 384x384 dimensions)
2. I'd like to know how this activation function is generalizable to the language autoregressive models. How next token diversity will vary with a context window.
3. Informative to see the loss curves to verify the convergence trend for models for 300 epochs (perplexity is a score)
4. In addition to the "Eskimo dog" class it would have been interesting to see evaluations for other examples also

**Questions:**

1. Please add GPT model details and citation
2. Table 2: It is worth noting that the IS for larger models is better with Softmax
3. Table 3,6: Put IS instead of Inception for consistency in a paper
4. line#431: 0.8% increase in GPU seconds per iteration (not 0.7%)
5. Table 5, right: I assume iteration is the epoch and not GPU seconds per training batch
6. line#425: typo

---

### Official Review · Reviewer_gJVn · 2024-11-03

**Soundness:** 2
**Presentation:** 3
**Contribution:** 3
**Rating:** 5
**Confidence:** 4

**Summary:**

This paper analyzes the negative impact of Softmax on current autoregressive generation tasks from the perspective of gradient bias in probabilistic activation functions. It points out that Softmax's excessive punishment of high-probability non-target classes affects the diversity of autoregressive generation tasks. At the same time, a new Gradient Suppressed Softmax (GS-Softmax) activation function is proposed, which reduces the gradient contribution to high-probability non-target classes. Ultimately, experiments demonstrate that this activation function enhances the diversity of generated content and optimization convergence.

**Strengths:**

1. This paper offers a new perspective, namely the gradient optimization of Softmax, to discuss the issue of impaired diversity in autoregressive generated content.
2. This paper proposes three criteria to guide the creation of probabilistic activation functions that conform to autoregressive characteristics, and introduces the corresponding Gradient Suppressed Softmax (GS-Softmax) function.
3. The writing style of this paper is clear, making it easy to understand.

**Weaknesses:**

1. This paper lacks analysis and discussion on an important issue. In the training of traditional large language models, the richness and diversity of training data can often mitigate the problem of over-punishment caused by Softmax. Does the improvement proposed in this paper work only when data is limited? This needs to be explored and experimentally verified.
2. The improvements in experimental results are too small. In all the experiments comparing Softmax and GS-Softmax, the enhancements in FID values are really minimal (for example, from 10.50 to 10.47), which seems to indicate that the impact of GS-Softmax on model performance is not critical or significant.
3. There is a lack of necessary experiments to support the motivation. Theoretically, the improvement of GS-Softmax on text should be more significant than the FID on images, which also supports the "rationality" maintained in the motivation. This experiment is important, yet the paper fails to provide it.

**Questions:**

## Justification For Recommendation And Suggestions For Rebuttal：
- Justification For Recommendation：Reference to Paper Strengths.
- Suggestions For Rebuttal:
  1. The analysis of experimental results needs to be more detailed.
  2. Present more experimental data to support the motivations behind the paper.

## Additional Comments For Authors：
To enhance clarity and persuasiveness, the authors should rectify vague descriptions and inaccuracies in the details.

---

### Official Review · Reviewer_uXfy · 2024-11-04

**Soundness:** 1
**Presentation:** 2
**Contribution:** 1
**Rating:** 3
**Confidence:** 4

**Summary:**

This paper starts from an observation that Softmax module in most models will over-penalize non-target classes with high prediction scores, which may be harmful to autoregressive tasks, since multiple valid predictions exist. To alleviate this, the authors propose Gradient Suppressed Softmax (GS-Softmax), which reduces the gradient contributions of high-probability non-target classes. Experimental results show that this proposed module improves the generation quality image tasks.

**Strengths:**

* The motivation and the proposed method is easy to follow.
* The proposed criteria for gradient suppressed softmax is general.

**Weaknesses:**

* The main weakness of the paper is the validity of the motivation. The authors claim that the softmax function will suppress possible valid non-target predictions. However, there are two concerns: 1) The authors do not provide any experimental evidence supporting this claim. During training or inference, does the model indeed generate only one high-probability prediction? 2) While the softmax function appears to suppress non-target predictions (e.g., $x_2, x_3, \dots, x_n$) based on its formulation, it is important to note that there may be instances in the dataset where other data shares the same conditional context with $x_2$ as the target. In such cases, $x_2$ would not be suppressed by the model.
* Additionally, the experimental results presented are insufficient to demonstrate the effectiveness of the proposed method. Improvements across all four evaluation metrics are generally no more than 5%, and the visualization results in Figure 2 do not show any clear superiority.
* There is a minor issue with the use of confusing brackets in Equation (5). It is recommended to use \left( and \right) in LaTeX to enhance clarity.

**Questions:**

* The paradigm of autoregressive generation originates from the NLP field. The motivation and method seems also valid in the NLP field. Would the authors consider conducting experiments on natural language tasks? Such experiments could enhance the generalizability of the proposed method.
* Are there any figures illustrating the training loss curve that could support the claims made in Section 5.3 regarding training efficiency?

---

### Note · Authors · 2024-11-25

**Comment:**

There is still room for improvement in this work, we will continue to make effort to enhance it.

**Withdrawal Confirmation:**

I have read and agree with the venue's withdrawal policy on behalf of myself and my co-authors.